

# Code4ML: a large-scale dataset of annotated Machine Learning code

Anastasia Drozdova[1], Ekaterina Trofimova[1], Polina Guseva[1], Anna Scherbakova[1] and Andrey Ustyuzhanin[1,2,3,4]

[1] Department of Computer Science, NRU Higher School of Economics, Moscow, Russia
[2] National University of Science and Technology MISIS, Moscow, Russia
[3] Constructor University, Bremen, Germany
[4] Institute for Functional Intelligent Materials, National University of Singapore, Singapore

## ABSTRACT

The use of program code as a data source is increasingly expanding among data scientists. The purpose of the usage varies from the semantic classification of code to the automatic generation of programs. However, the machine learning model application is somewhat limited without annotating the code snippets. To address the lack of annotated datasets, we present the Code4ML *corpus*. It contains code snippets, task summaries, competitions, and dataset descriptions publicly available from Kaggle—the leading platform for hosting data science competitions. The *corpus* consists of ~2.5 million snippets of ML code collected from ~100 thousand Jupyter notebooks. A representative fraction of the snippets is annotated by human assessors through a user-friendly interface specially designed for that purpose. Code4ML dataset can help address a number of software engineering or data science challenges through a data-driven approach. For example, it can be helpful for semantic code classification, code auto-completion, and code generation for an ML task specified in natural language.

## INTRODUCTION

In recent years, more and more tools for software development have started using machine learning (ML) (*Allamanis et al., 2018*; *Yang et al., 2021*). ML systems are capable of analyzing (*Alsolai & Roper, 2020*; *Bilgin et al., 2020*), manipulating (*Goues, Pradel & Roychoudhury, 2019*; *Liu et al., 2020*), and synthesizing (*Svyatkovskiy et al., 2020*; *Austin et al., 2021*) code. However, even the most successful deep-learning models of the last few years (*Roziere et al., 2020*; *Chen et al., 2021*) require training on vast amounts of data before obtaining good results.

There is a multitude of code datasets (*Iyer et al., 2018*; *Puri et al., 2021*). Still, most of them need to be domain-specific, which poses a challenge during the development of tools for specialized areas of software engineering because of domain shift (*Gretton et al., 2006*). Moreover, generic datasets can lack examples, making it hard for the model to pick up on domain-specific patterns.

ML is one of the most popular software development areas without a domain-specific code *corpus*. Such a dataset is necessary for the development of data science tools. The searchable database of annotated code is suitable for data scientists to find solutions to

Corresponding author
Ekaterina Trofimova,
etrofimova@hse.ru

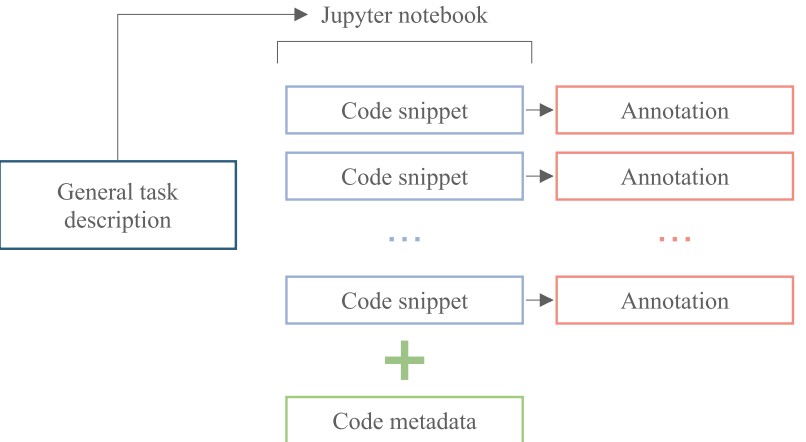

**Figure 1 The scheme of the ML code *corpus* candidate.**

specific problems they are working on. This can be especially useful if the code is organized and tagged in a way that makes it easy to find relevant examples. The desired structure of the ML code *corpus* is illustrated by Fig. 1. Such a *corpus* allows researchers to train ML models to predict the correct code for a given data science problem. One can use the annotated code as training data to build a model that takes in a description of a problem and generates code that solves it.

Also, the possible application of the annotated code lies in developing tutorials or educational materials for data scientists. By providing explanations and examples of how to solve real-world problems, one can help others learn and improve their skills.

Overall, a *corpus* of annotated code, accompanied by a natural task description, can be a valuable resource for data scientists and others working in applied data science.

In this article, we introduce a Large-scale Dataset of Machine Learning Code (Code4ML) dataset, a *corpus* of Python code snippets, competition, and data summaries from Kaggle.

Our major contributions are the following:

- We present a large dataset of about 2.5 million Python code snippets from public Kaggle notebooks. Those snippets are enriched with metadata.
- The notebooks are accompanied by natural language descriptions of corresponding competitions.
- We propose a novel technique for ML code annotation based on a Machine Learning Taxonomy Tree that reflects the main steps of the ML pipeline. In addition, we provide an annotation tool that can help continue further markup of the dataset.

The rest of this article is organized as follows. "Related Work" contains an overview of existing datasets and their properties. "Construction of Code4ML" includes a description of our dataset collection/annotation process. Details of human and machine-oriented reading of the dataset are described in "Code4ML Dataset Structure". "Downstream Tasks" describes potential applications and research directions that the community can perform

**Table 1 Overview of some of the existing ML-related datasets for Python.**

| Dataset name | Dataset size | Human-curated annotated data size | Data source | Natural description of the general task the code is written for |
|---|---|---|---|---|
| Boa *Biswas et al. (2019)* | ≈5M Python files | – | GitHub | – |
| JuICe *Agashe, Iyer & Zettlemoyer (2019)* | ≈1.5M code snippets | ≈4K code snippets | GitHub | – |
| KGTorrent *Quaranta, Calefato & Lanubile (2021)* | ≈250K Jupyter notebook files | – | Kaggle | – |
| Code4ML (ours) | ≈2.5M code snippets | ≈8K unique code snippets | Kaggle | ✓ |

with the presented dataset. "Limitations" reflects the limitations of the *corpus*. "Conclusion" concludes the article.

## RELATED WORK

Several publicly available datasets for source code have been proposed for various code intelligence tasks. Some datasets, like CodeNet (*Puri et al., 2021*) and POLYCODER's dataset (*Xu et al., 2022*), contain snippets from different programming languages. Others consist of code from one specific language: PY150 (*Raychev, Bielik & Vechev, 2016*) for Python, CONCODE (*Iyer et al., 2018*) for Java, Spider (*Yu et al., 2018*) for SQL, *etc*. The source code is collected from GitHub (CodeSearchNet *Husain et al., 2019*) and Stack Overflow (CoNaLa *Yin et al., 2018*), and from other platforms as well, such as Kaggle (*Quaranta, Calefato & Lanubile, 2021*). In *Lu et al. (2021)* CodeXGLUE is proposed, a machine learning benchmark dataset that contains 14 datasets of different sizes and in multiple programming languages.

Table 1 gives an overview of several datasets for Python since our *corpus* is also for Python. As we aim to study ML code, we focus on ML-related datasets.

### Boa

The Boa (*Biswas et al., 2019*) dataset represents a pool of data-science-related python files and the meta-information about the corresponding GitHub projects. The authors extract the abstract syntax tree (AST), *i.e.*, a tree representation of a conceptual code structure, from the source code and store AST parts classified into domain-specific types: ASTRoot, containing a program file, namespace, holding the qualitative path to the file; declarations, including functions as methods in Python, which in turn have other statements and expressions. While maintaining the project as a repository containing different program files remains the standard among the Data Science community, interactive programming in Jupyter notebooks is gaining popularity. Moreover, Jupyter usually provides a logical division of code into snippets. That makes it possible to analyze the ML pipeline's structure quickly, *e.g.*, a code snippet corresponding to the data import is further followed by data processing and model training.

### JuICe

In *Agashe, Iyer & Zettlemoyer (2019)*, the authors provide the set of manually created high-quality Jupyter notebooks representing class programming assignments. The notebooks

**NL:** Load features and labels in a dataframe.

```python
import pandas as pd
X = pd.read_json('features.json')
y = pd.read_json('labels.json')
```

**NL:** Split the data into train and test.

```python
from sklearn.model_selection import train_test_split
X_train, X_test, y_train, y_test = train_test_split(X, y)
```

**NL:** Create and train the model.

```python
from sklearn.tree import DecisionTreeClassifier
dtree = DecisionTreeClassifier()
dtree.fit(X_train, y_train)
```

**Figure 2** **JuICE code snippets with the corresponding natural language description examples.** Source: *Agashe, Iyer & Zettlemoyer (2019)*. 

consist of alternating NL markdown and code cells. The code is assumed to match the provided markdown description. The *corpus* includes 1.5 million unique target cell-context examples. A human-curated test part of 3.7 K examples is also provided. The motivation of the JuICe dataset lies in the generation of the code snippet by the natural description of the Jupyter notebook cell using the prior information from the notebook. However, the description of the task the notebook tries to solve needs to be included. Thus, JuICe is hard to use to solve the problem of ML pipeline generation.

### KGTorrent

In *Quaranta, Calefato & Lanubile (2021)* the authors present a KGTorrent dataset. It includes a complete snapshot of publicly available artifacts of Kaggle, including Jupyter notebooks, dataset descriptions, and forum discussions. Still, there are no descriptions of the competitions. Also, as KGTorrent only aggregates the Kaggle data, it does not includes any specific annotation of the code snippets. Moreover, one can verify Kaggle notebook quality by assessing the Kaggle score of the notebook, which corresponds to the value of the specified competition metric. Because Kaggle competitions have various metrics, code quality assessment is difficult. Thus, although KGTorrent is an extensive collection of the Jupyter notebooks and Kaggle metadata, it is unsuitable for ML pipeline synthesis from natural language description.

## CONSTRUCTION OF CODE4ML

Our work focuses on the Kaggle kernels (Jupyter Notebooks) as the sequential computational code cells designed to solve machine learning problems. We aim to reduce the dimension of the learning space by introducing a taxonomy tree once it is used as an annotation class for notebook code cells. One can compare this annotation with the markdown describing the task of the code cell in the JuICe dataset (see Figs. 2 and 3). Unlike markdown-based annotation, our taxonomy class approach is uniquely defined in all snippets. We provide a set of $\approx 8K$ human-curated annotated unique code snippets and a tool for the snippets' manual classification. Thus, the human assessors describe the whole

**Taxonomy type:** Data_extraction.load_from_disk

```
X = pd.read_json('features.json')
y = pd.read_json('labels.json')
```

**Taxonomy type:** Data_transform.prepare_x and_y

```
X_train, X_test, y_train, y_test = train_test_split(X, y)
```

**Taxonomy type:** Model_train.choose_model_class

```
dtree = DecisionTreeClassifier()
```

**Taxonomy type** Model_train.train_model

```
dtree.fit(X_train, y_train)
```

**Figure 3 Code4ML code snippets with the corresponding taxonomy types examples.**

ML pipeline, *i.e.*, the sequence of the taxonomy tree vertices. Like KGTorrent, our *corpus* also contains information about Kaggle notebooks, corresponding datasets, and competitions. We deal with the problem of kernel verification by introducing the classification of Kaggle metrics into 20 classes. Moreover, each competition in the *corpus* is provided with a natural description.

## Collection and preprocessing of the dataset

Kaggle is the most prominent platform for competitive data science. It curates the creation of data challenges and encourages users to publish their solutions in Jupyter Notebook kernels. A kernel is a sequence of code snippets and description blocks in a natural language. The code from the kernels is a good source of ML code since the users have to build their machine learning pipelines from scratch.

Kaggle provides an API for accessing the published kernels and competitions and an open dataset containing various metadata. Using the API, we collect the most popular kernels from the most popular competitions (*i.e.*, with the highest number of teams). We only consider kernels that use Python3 and have Apache 2.0 license.

The parser processes the collected kernels for code blocks and corresponding kernel id extraction. Each code cell of the Jupyter notebook is considered a code snippet. We clean it up to ensure the collected code uniformity by removing broken Unicode characters and formatting the code to conform to the PEP8 standard. Also, personal information such as emails is not included in the snippets.

Notebooks on Kaggle have many useful metrics. Users vote for notebooks with high-quality code. Another important notebook metric is the kernel result on the test set (Kaggle score).

This metadata, as well as a number of kernel comments, are collected from Meta Kaggle.[1]

## Taxonomy tree

Transformation of the Python code into conceptual pipelines describing the steps for performing ML experiments significantly reduces the amount of data required to train an

[1] Kaggle's public data on competitions, users, submission scores, and kernels (meta-kaggle).

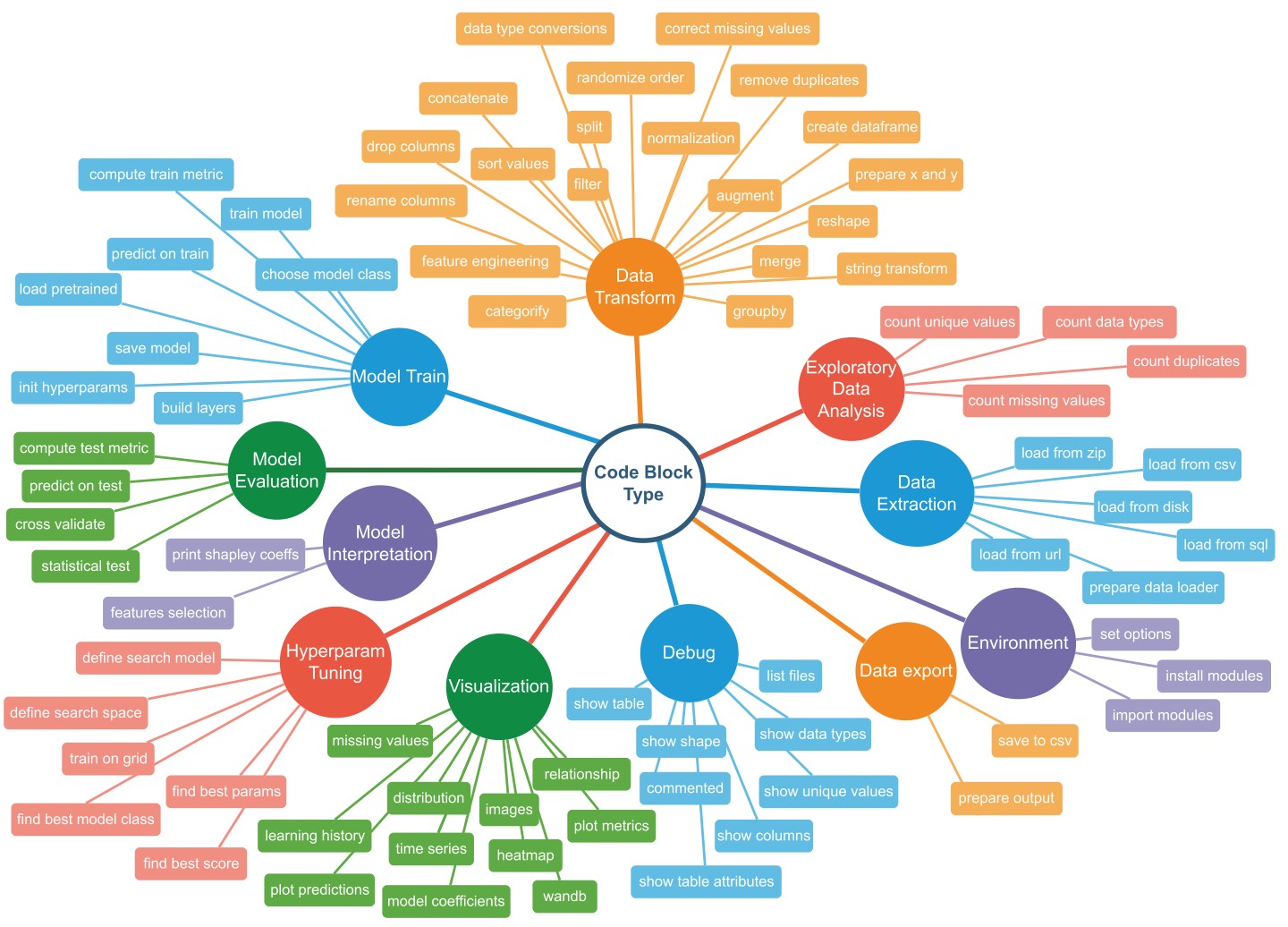

**Figure 4  Machine learning taxonomy tree.**

ML model to analyse or generate the sequence. Almost any Jupyter Notebook can be translated into such a pipeline of repeating typical patterns.

To describe code blocks from a notebook, we have developed a set of categories and combined them in a Taxonomy Tree. The tree has two levels: the upper level denotes a high-level classification of an ML pipeline step. Each descendent vertex corresponds to a more specific action. The second-level vertices are called *semantic types*. So, for example, semantic type `mising values` in `Visualisation` category represents an action of displaying missing values properties, such as quantities *vs.* features. In contrast, `correct missing values` in `Data Transform` represents filling it with a default value or removing the rows with missing values completely. There are 11 upper-level categories and ≈80 lower-level classes. Figure 4 illustrates the graph. Figure 5 shows examples of code snippets corresponding to different graph vertices.

| Semantic class | Example |
|---|---|
| Data_Transform.drop_column | ```train_df.drop("Date", inplace=True, axis=1)```<br>```test_df.drop("Date", inplace=True, axis=1)``` |
| Model_Train.choose_model_class | ```from sklearn import linear_model```<br><br>```reg_CC = linear_model.Lasso(alpha=0.1)```<br>```reg_Fat = linear_model.Lasso(alpha=0.1)``` |
| Hyperparams.define_search_space | ```parameters = {'lstm_nodes': [14,16,20],```<br>```              'nb_epoch': [50],```<br>```              'batch_size': [32],```<br>```              'optimizer': ['adam']}``` |
| Visualization.distribution | ```fig = plt.figure()```<br>```fig.suptitle("Algorithm_Comparison")```<br>```ax = fig.add_subplot(111)```<br>```plt.boxplot(results)```<br>```ax.set_xticklabels(names)```<br>```plt.show()``` |
| Data_Transform.normalization | ```scaler = MinMaxScaler()```<br>```df['revenue'] = scaler.fit_transform(```<br>```    df[['revenue']]```<br>```)``` |
| Model_Train.train_model | ```rfr = RandomForestRegressor(```<br>```    n_estimators=200,```<br>```    max_depth=5,```<br>```    max_features=0.5,```<br>```    random_state=449,```<br>```    n_jobs=-1```<br>```)```<br>```rfr.fit(x_train, y_train)``` |

**Figure 5** Semantic typification of code snippets example.

Creating the ML Taxonomy Tree relies on data science standards such as CRISP-DM (*Shearer, 2000*) and ISOTR24029 (*ISO/IEC TR 24029-1:2021, 2021*), the experts' experience in machine learning and data science.

# CODE4ML DATASET STRUCTURE

The data is organized as a set of tables in CSV format. It includes several central entities: raw code blocks collected from Kaggle (`code_blocks.csv`), kernels (`kernels_meta.csv`) and competitions meta information (`competitions_meta.csv`). Annotated code blocks are presented in a separate table `markup_data.csv`. Each code block is associated with a semantic type assigned to it by an external assessor. A dictionary of semantic types is stored in the table `vertices.csv`.

Code snippets information (`code_blocks.csv`) can be mapped with kernels metadata *via* `kernel_id`. Kernels metadata is linked to Kaggle competitions information through `comp_name` (Fig. 6). To ensure the quality of the data `kernels_meta.csv` includes only

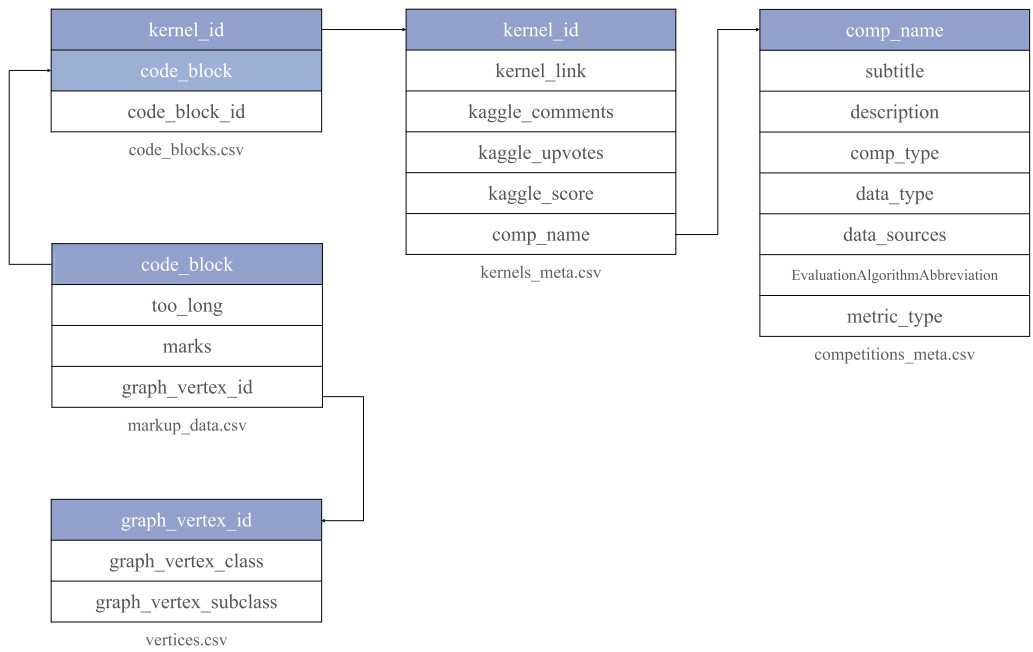

**Figure 6 Code4ML corpus structure.** Each table is stored in a separate file with a unique key. It is highlighted on the figure and used to reference its entries outside.

Jupyter Notebooks with a non-empty Kaggle score. The data is published online at the Zenodo platform (*Drozdova et al., 2022*).

Each *competition* entry has the text description and metadata, reflecting competition, dataset characteristics, and evaluation metrics. `EvaluationAlgorithmAbbreviation` is collected from Meta Kaggle and provides additional information on competitions and notebooks. `EvaluationAlgorithmAbbreviation` has 92 unique values, which make it difficult to filter the kernels by scores concerning the metric. To tackle it, we group `EvaluationAlgorithmAbbreviation` into 20 classes reflected in the `metric_type` column. Figure 7 shows the distribution of the `metric_type`. The class description is provided in Fig. 8.

The dataset for the corresponding competitions can be downloaded using Kaggle API: `kaggle competitions download -c data_source`, where `data_source` is the name of the dataset at Kaggle.

The *code_blocks* entry includes the code snippet, the corresponding kernel id, and the code block id, which is the index number of the snippet in the Jupyter Notebook.

The *corpus* contains 107,524 notebooks. Most of those (23,104) are assigned to competitions. Thus, 625,125 snippets belonging to those notebooks have a kernel score value.

We use a web form for manual sorting of code snippets into semantic classes[2]. The form allows marking code snippets according to their semantic type described in "Construction of Code4ML" as well as cleanliness and the kind of data (*i.e.*, table, image, *etc*.) To specify the markup confidence level in the resulting class, one should choose the corresponding

[2] Additional labeled data is always welcome. You can participate at https://nl2ml-form.coresearch.club/. Please keep in mind that the registration of the assessor needs to be approved by the Code4ML team members.

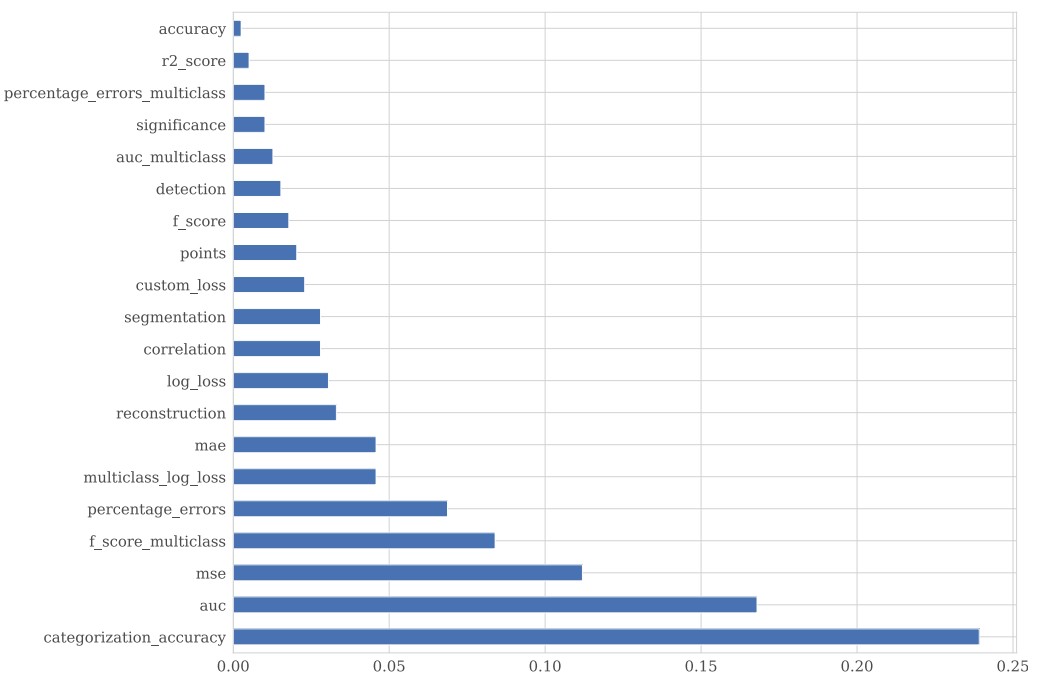

**Figure 7 Distribution of the competition's metric type.**

value of marks (from 1 to 5). The *too_long* flag denotes the purity of the snippet to be marked up. The flag should be set if the cell code can not be unambiguously attributed to a single semantic type, *i.e.*, it contains many different semantic types. The detailed markup rules are in Fig. 9. `markup_data.csv` includes data labeled by the Code4ML project team. The interface of the web form is shown in Fig. 10. All assessors must follow the markup rules.

The *markup* table contains the following fields: the id of the parent notebook, code snippet text, the boolean *too_long* flag, the assessment confidence score in the range from 1 to 5 (best), and the id of the snippet class chosen by the assessor.

In total, assessors marked around 10,000 snippets (some snippets are similar across notebooks, after that, there are ≈8,000 unique snippets). A total of ≈68% of marked snippets got the highest confidence score (*i.e.*, 5), while ≈18% and ≈11% got the confidence score equal to 4 and 3, correspondingly.

To annotate the rest of the *corpus*, we provide the general assessment of the automatic code snippets labeling.

We use the manually labeled code snippets for training the basic models. The class distribution of the snippets can be found in Fig. 11. We report two metrics: accuracy and F1-score.

Since the code block is a sequence of symbols, an encoding is required. We used frequency-inverse document frequency (*Papineni, 2001*) as a vectorizer.

We use support vector machines (SVM) (*Boser, Guyon & Vapnik, 1992*) based models for snippets classification. This method does not require much data for training, so this approach is used as a reference ML method. We apply SVM with different kernels: linear,

| Class name | Description | Aim |
|---|---|---|
| accuracy | the share of the correct answers | maximisation |
| r2_score | coefficient of determination | maximisation |
| percentage_errors_multiclass | multiclass classification percentage error | minimisation |
| significance | custom metrics reflecting the predictions certainty | maximisation |
| auc_multiclass | generalization of ROCAUC to a multiclass classification | maximisation |
| detection | object detection metrics | maximisation |
| f_score | $F_\beta$-score metrics | maximisation |
| points | reinforcement learning metrics | maximisation |
| custom_loss | custom loss metrics | minimisation |
| segmentation | objects segmentation metrics | maximisation |
| correlation | correlation metrics | maximisation |
| log_loss | logarithmic loss | minimisation |
| reconstruction | reconstruction metrics | maximisation |
| mae | mean average error metrics, e.g. WMAE | minimisation |
| multiclass_log_loss | logarithmic loss generalisation to multiclass classification | minimisation |
| persentage_errors | percentage error metrics, e.g. RMSLE, mape | minimisation |
| f_score_multiclass | generalisation of f_score to multiclass classification problems | maximisation |
| mse | mean squared error metrics, e.g. mse, RMSE | minimisation |
| auc | ROCAUC | maximisation |
| categorization_accuracy | generalisation of accuracy to multiclass classification problems | maximisation |

**Figure 8  Characterisation of metric type classes.**

polynomial, and Radial Basis Function (RBF). The hyperparameters are selected based on cross-validation metrics on ten folds. The multiclass case is handled using a one-*vs*-all scheme (*Chang & Lin, 2011*). Details of the model training are available in Fig. 12.

Figure 13 illustrates the level of similarity between the manually assessed sample and the whole data. This plot shows the cumulative distribution function for the labeled and the total samples. The horizontal axis shows the prediction of a calibrated SVM classifier with a linear kernel trained on 80% of the labeled data. The probability ratio of the classes predicted by the model that does not exceed the specified threshold is then compared for the test part of the markup data (orange line) and the entire `code_blocks.csv` table (blue

While marking up the data using the web form, one should take into account the following suggestions:

- If code from only one semantic type is found in the snippet, *mark* 5 is to be set;

- If the cell code can not be unambiguously interpreted, the *too_long* flag should be set up;

- If the *too_long* flag flag is set, the maximum possible *mark* is equal to 4 (thus, the *too_long* flag and confidence 5 can not be set at the same time);

- If the snippet contains most of the code of one semantic type, but there is code of other types, then the type to which the most of the code belongs should be set with *mark* equal to 4;

- If the snippet shares the same amount of code of different semantic types, the type that comes first with *mark* equal to 3 should be set;

- If a sequence of operations for which semantic types are defined is applied to the data (e.g., sequence of *concatenate* and *groupby* methods) in one raw of code, then the semantic type of such a snippet will be the type of the last applied operation.

**Figure 9** **Manual code snippets labeling algorithm.**

**Figure 10** **Interface of the WEB form.** The web form allows the users to annotate the code snippets. On the left there is an example of code snippet as well as the link to the original Kaggle kernel. On the right there are fields for manual labeling. Due to a large amount of options, the selection of semantic class is split into two parts.

line). Although the data in the whole dataset is not identical to the labeled data, one can see the closeness of the two lines, which allows us to conclude that the labeled sample is moderately representative.

The semi-supervised models (*Xie et al., 2020*) for the snippets classification are applied to deal with the lack of manually labeled data.

First, a linear kernel-based SVM model is trained on the marked-up dataset. We collect the prediction of the trained model on the unlabeled part of the data. The predictions are further used as pseudo labels in combination with marked-up data to train a different SVM model with the RBF kernel. The results can be found in Table 2.

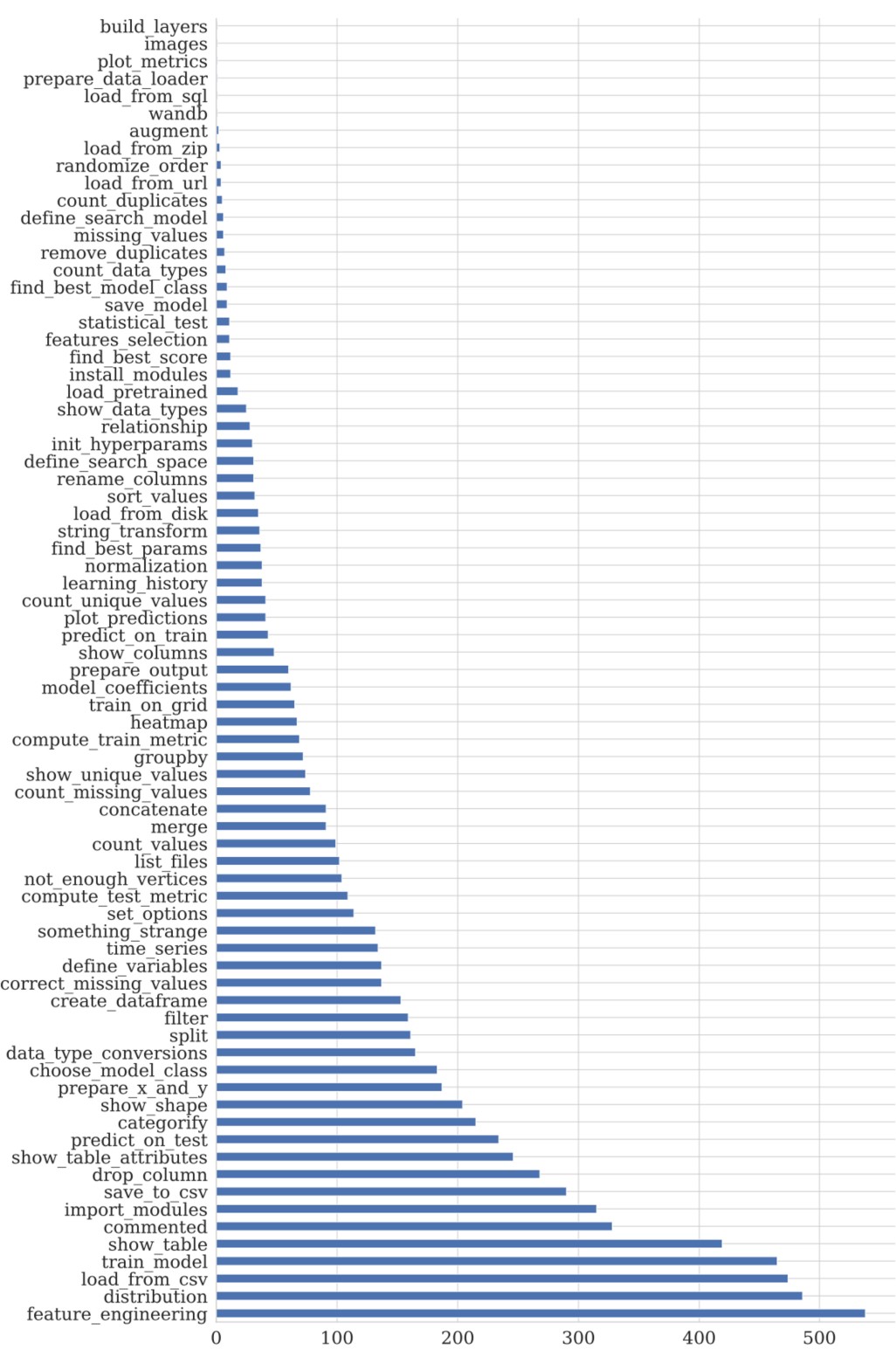

**Figure 11 Distribution of the markup data taxonomy type.** The dominated data type of the corresponding to the markup snippets competitions datasets is tabular. That leads to the imbalance in the semantic class distribution.

| Model | Hyperparameter | Type | Value |
|---|---|---|---|
| SVM + Linear | C | numeric | 37.17 |
| | min_df for TF-IDF | integer | 2 |
| | max_df for TF-IDF | numeric | 0.31 |
| SVM + Poly | C | numeric | 1.43 |
| | Degree of poly kernel | integer | 3 |
| | min_df for TF-IDF | integer | 6 |
| | max_df for TF-IDF | numeric | 0.30 |
| SVM + RBF | C | numeric | 8.71 |
| | min_df for TF-IDF | integer | 7 |
| | max_df for TF-IDF | numeric | 0.39 |
| Pseudo labels 20% | C | numeric | 98.37 |
| | Kernel type | categorical | linear |
| | min_df for TF-IDF | integer | 3 |
| | max_df for TF-IDF | numeric | 0.53 |
| Pseudo labels 40% | C | numeric | 121.59 |
| | Kernel type | categorical | linear |
| | min_df for TF-IDF | integer | 3 |
| | max_df for TF-IDF | numeric | 0.41 |
| Pseudo labels 100% | C | numeric | 145.56 |
| | Kernel type | categorical | linear |
| | min_df for TF-IDF | integer | 2 |
| | max_df for TF-IDF | numeric | 0.26 |

**Figure 12 The resulted hyperparameters for automatic snippets classification model.** The hyperparameters for SVM models are selected by cross-validation on ten folds using *Akiba et al. (2019)*. The kernel can be Linear, Poly or RBF. The regularization parameter C is selected from [0.1, 1000].

## DOWNSTREAM TASKS

The proposed *corpus* of the publicly available Kaggle code snippets, task summaries, competitions, and dataset descriptions publicly enriched with annotation and useful metadata is a valuable asset for various data-driven scientific endeavors.

### ML code classification

As shown above, one can use Code4ML for a semantic code classification task, where each ML code snippet should be labeled as one of the taxonomy tree classes illustrated by Fig. 4. This task helps to summarize ML pipelines. One can use the proposed baseline models as a starting point for the semantic ML code classification.

### ML pipeline synthesis

The availability of the Kaggle competition description and the markup data makes training NL to ML code generative models possible. As mentioned earlier, the implementation of data analysis pipelines usually comes down to building a combination of repeating typical patterns. Nevertheless, constructing such pipelines is a crucial skill for specialists in various

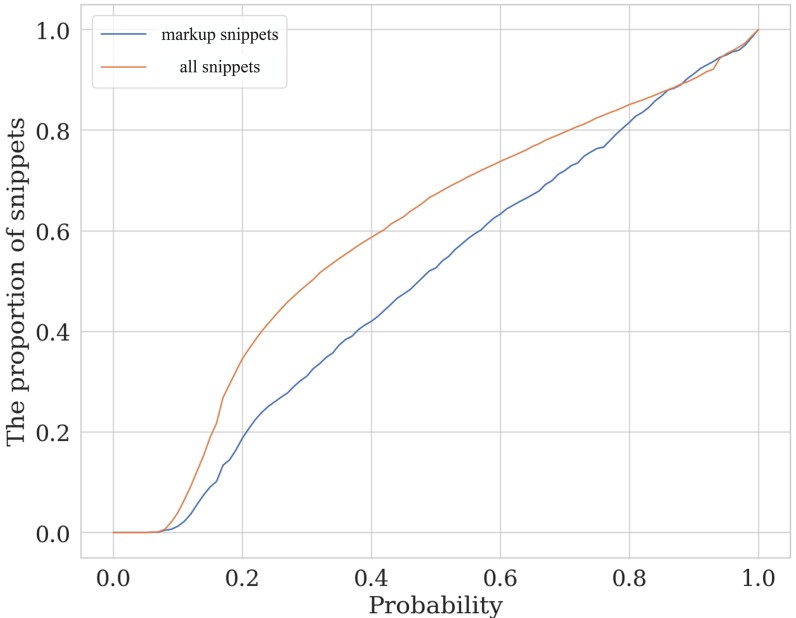

**Figure 13 The valuation of the similarity of assessed and unassessed snippets.** The plot lines show the cumulative distribution function (CDF) for the labeled (markup) and full (all snippets) samples depending on semantic class predicted probability.

**Table 2 Ten-folds cross-validation performance of the baseline models for automatic data labeling.**

|  | Metrics | |
| --- | --- | --- |
| **Model** | **F1-score** | **Accuracy** |
| SVM + Linear | 0.684 ± 0.024 | 0.691 ± 0.022 |
| SVM + Poly | 0.619 ± 0.021 | 0.625 ± 0.019 |
| SVM + RBF | 0.689 ± 0.022 | 0.625 ± 0.019 |
| SVM with 20% of pseudo labels | 0.831 ± 0.014 | 0.834 ± 0.014 |
| SVM with 40% of pseudo labels | 0.845 ± 0.016 | 0.851 ± 0.014 |
| SVM with 100% of pseudo labels | **0.872** ± 0.004 | **0.872** ± 0.004 |

**Note:**
The best results are in bold.

subject areas that are not directly related to data analysis but rely on such competencies in some works. An example would be biologists, chemists, physicists, or representatives of the humanities. Code4ML provides the data for translating tasks described in a natural language into a programming language (Python). The annotation of the code, gained by snippets classification to semantic types, can serve as additional information or control input for the code generation model.

## ML code auto-completion

Code4ML also covers a lack of annotated data for ML code auto-completion. Code completion is the most popular software development technique (*Murphy, Kersten & Findlater, 2006*) and is found in every powerful IDE. It can be used as a typing assistant tool for discovering relevant libraries and APIs.

Nevertheless, most existing code completion systems fail on uncommon completions despite their importance for real-world efficacy (*Hellendoorn et al., 2019*). Training a code completion model on domain-specific data can help determine the too-rare patterns in generic code datasets and improve real-world accuracy.

## LIMITATIONS

There are a few potential limitations and risks to consider when working with a *corpus* of annotated code.

The *corpus* may only contain code for specific problems or a limited range of programming languages, making it less useful for other types of data science work. The *corpus* may be biased regarding the kinds of problems it covers or the approaches used to solve them. This could lead to incomplete or misleading information if the *corpus* does not represent the full range of problems and techniques used in data science.

Code4ML aggregates the most popular Kaggle competitions. This ML contest platform divides challenges into several types. Community prediction competitions, full-scale machine learning problems, relatively simple ML tasks, and more experimental (research) issues are the most common. The Code4ML shares of these problem types are 50%, 25%, 11%, and 10%, respectively. The majority of the aggregated competitions operate with the table data (52%), image (28%), and text (11%) data. Reinforcement learning or audio/video processing problems are less widespread.

Depending on the source of the annotated code, the quality of the annotations and the code itself may vary. It is important to carefully evaluate the credibility and accuracy of the annotations to ensure that the *corpus* is reliable and valuable. We mitigate this threat by providing the general confidence assessment and the ratio of ambitiousness for each snippet annotation.

### Legal considerations

Legal risks may be associated with using code from the *corpus* in the projects. It's vital to ensure one has the necessary permissions and licenses to use the code and adequately attribute any code used.

Overall, it's essential to carefully consider the limitations and risks of a *corpus* of annotated code usage and to make sure that the researcher uses it in a way that is ethical, legal, and useful for his specific needs. The *corpus* is published under Creative Commons Attribution 4.0 International license.

## CONCLUSION

This article describes a novel Large-scale Dataset of annotated Machine Learning Code (Code4ML) containing ML code snippets in Python and corresponding ML tasks metadata.

The dataset contains problem descriptions from ≈400 Kaggle competitions in natural language. It also includes more than 20 thousand public Python 3 notebooks representing machine learning pipelines for solving those competitions with the provided Kaggle score.

Those notebooks comprise around ≈600 thousand code cells. We propose a taxonomy graph to describe the code snippets as principal parts of the ML pipeline.

The current version of the dataset only covers part of the scope of Kaggle ML code snippets, and it can be easily extended in the future.

Around ten thousand snippets have been manually labeled to date. We developed a data markup web application that can help volunteers contribute to the extension of the markup dataset and eventually cover it entirely. Consequently, we warmly welcome any efforts from the community in this direction.

We are confident that the Code4ML dataset can be helpful for various vital modern ML challenges, such as code classification, segmentation, generation, and auto-completion. Hopefully, it can also open up new venues for AutoML research.

## ACKNOWLEDGEMENTS

We want to acknowledge the considerable time and efforts spent annotating the Code4ML *corpus* by Alexander Levin, Ivan Pyaternev, Marina Stepanova, Valery Berezovskiy, Evgenia Yegorova, Julia Gorshkova, Anastasia Gorodilova, Aynur Nureyev, Anastasia Denisenko, Maria Akimenkova, Daria Sapozhnikova, Alexander Myltsev.

### Funding

The publication was supported by the grant for research centers in the field of AI provided by the Analytical Center for the Government of the Russian Federation (ACRF) in accordance with the agreement on the provision of subsidies (identifier of the agreement 000000D730321P5Q0002) and the agreement with HSE University No. 70-2021-00139. The funders had no role in study design, data collection and analysis, decision to publish, or preparation of the manuscript.

### Grant Disclosures

The following grant information was disclosed by the authors:
Analytical Center for the Government of the Russian Federation (ACRF): 000000D730321P5Q0002.
HSE University: 70-2021-00139.

### Competing Interests

The authors declare that they have no competing interests.

### Author Contributions

- Anastasia Drozdova conceived and designed the experiments, performed the experiments, performed the computation work, prepared figures and/or tables, authored or reviewed drafts of the article, and approved the final draft.
- Ekaterina Trofimova conceived and designed the experiments, performed the experiments, analyzed the data, performed the computation work, prepared figures and/ or tables, authored or reviewed drafts of the article, and approved the final draft.

- Polina Guseva conceived and designed the experiments, performed the experiments, analyzed the data, performed the computation work, authored or reviewed drafts of the article, and approved the final draft.
- Anna Scherbakova analyzed the data, authored or reviewed drafts of the article, and approved the final draft.
- Andrey Ustyuzhanin conceived and designed the experiments, authored or reviewed drafts of the article, and approved the final draft.

## Data Availability

The Code4ML: a Large-scale Dataset of annotated Machine Learning Code (1.0.1) is available at Zenodo: Anastasia Drozdova, Polina Guseva, Ekaterina Trofimova, Anna Scherbakov, Andrey Ustyuzhanin, Anastasia Gorodilova, & Valery Berezovsky. (2022). Code4ML: a Large-scale Dataset of annotated Machine Learning Code (1.0.0) [Data set]. Zenodo. https://doi.org/10.5281/zenodo.7023415.

The source code for the data collection is available at GitHub: Ekaterina Trofimova. (2022). ketrint/Code4ML: v1.0.0 (v1.0.0). Zenodo. https://doi.org/10.5281/zenodo.7144838.

The code for the classification experiments is available at GitHub: Anastasia Drozdova. (2022). ADrozdova/NL2ML-code-classification: v1.0 (v1.0). Zenodo. https://doi.org/10.5281/zenodo.7144858.

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
