# Peer review of "Code4ML: a large-scale dataset of annotated Machine Learning code"

_PeerJ Computer Science, doi:10.7717/peerj-cs.1230_

## Round 0.1 · original submission · Minor Revisions

The suggested changes from the reviewers are presentation issues. Please improve the following:
- Describe and summarise more clearly the comparison to existing data sets
- Strengthen the motivation for creating and publishing a new data set
- Discuss further uses of the data set
- Discuss in more detail the limitations of the data set

Reviewer 1 ·

Basic reporting

The research background is not enough. The authors should add stronger research motivation for contributing a new dataset.

Experimental design

The knowledge gap between the existing datasets and this new dataset should be summarized and added in the Introduction section.

Validity of the findings

The authors should enhance the impact of their proposed dataset by pointing out more potential applications as the future research directions, as specific as possible, e.g., with input and output description.

Additional comments

Dear authors, overall this is a good study. My concerns are about two questions:
1. Why is a new dataset needed?
2. How could this new dataset be used in the future?
For the first question, the authors may want to add more discussion in the Introduction section. For the second question, the authors may want to add more discussion in the DOWNSTREAM TASKS section.

Reviewer 2 ·

Basic reporting

The author introduced a novel technique for ML code annotation based on a Machine Learning Taxonomy Tree that reflects the main steps of the ML pipeline. Additionally, the author provides an annotation tool that can help continue further markup of the dataset.
From a journal perspective, the contribution is minor.

Experimental design

This paper is focused on a large-scale dataset of annotated Machine Learning code. More technical revisions are required in this paper about the Algorithm and Flowchart of the proposed methodology. The experimental section needs to be clarified.

Validity of the findings

Data and Analysis experimentation-based findings need to be revised. For example, different validation techniques need to be applied. Also, the author needs to show a comparison with these schemes.

Additional comments

The Paper needs the following Major Revisions:
• The limitations of the existing methods proposed for similar tasks should be pointed out and explained in how the proposed strategy would address each limitation.
• The results are better. But from a technical perspective, it is mainly a combination of existing technologies, and the contributions could be more evident. The author needs to highlight the contributions of this paper in the revised version.

·

Basic reporting

- Language is clear and unambiguous
- Literature can be improved a bit.
- Structure, facts and figures are apt, no changes required.
- Results are sound enough.

Experimental design

Not changes required.

Validity of the findings

Good enough, no changes required.

---

## Round 0.2 · accepted · Accept

I checked all reviewer comments in detail and you have improved the article accordingly. Well done! The paper is now ready for publication.